# A Novel Strain Breeding of *Ganoderma lucidum* UV119 (Agaricomycetes) with High Spores Yield and Strong Resistant Ability to Other Microbes’ Invasions

**DOI:** 10.3390/foods12030465

**Published:** 2023-01-19

**Authors:** Chuanhong Tang, Yi Tan, Jingsong Zhang, Shuai Zhou, Yoichi Honda, Henan Zhang

**Affiliations:** 1National Engineering Research Center of Edible Fungi, Key Laboratory for the Utilization of Edible Fungi in Southern China, Ministry of Agriculture, Institute of Edible Fungi, Shanghai Academy of Agricultural Sciences, Shanghai Key Open Laboratory of Agricultural Genetic Breeding, Shanghai 201403, China; 2Laboratory of Forest Biochemistry, Graduate School of Agriculture, Kyoto University, Kyoto 6068502, Japan

**Keywords:** *Ganoderma lucidum*, protoplasts, mutagenic breeding, basidiospore yield

## Abstract

The spore powder of *Ganoderma lucidum* (*G. lucidum*) has been proven to have a variety of pharmacological activities, and it has become a new resource for the development of health products and pharmaceuticals. However, the scarcity of natural resources, strict growth conditions and difficulty in controlling the stable yield, and quality of different culture batches seriously limit the development and utilization of *G. lucidum* spore powder. In the present study, the strain with the highest spore powder yield, G0109, was selected as the original strain to generate mutants of *G. lucidum* using ultraviolet ray irradiation. A total of 165 mutagenic strains were obtained, and fifty-five strains were chosen for the cultivation test. Importantly, one mutagenic strain with high spore powder yield and strong resistance to undesired microorganisms was acquired and named strain UV119. More cultivations demonstrated that the fruiting body and basidiospore yields from UV119 were, respectively, 8.67% and 19.27% higher than those of the parent (G0109), and the basidiospore yield was 20.56% higher than that of the current main cultivar “Longzhi No.1”. In conclusion, this study suggested that ultraviolet ray irradiation is an efficient and practical method for Ganoderma strain improvement and thus provided a basis for the development and application of *G. lucidum* spore production and outstanding contributions to the rapid development of the *G. lucidum* industry.

## 1. Introduction

*Ganoderma lucidum* (Curtis) P. Karst, called a “top Chinese medicinal material” in *Shennong’s Classic of Meteria Medica*, is a treasure of traditional Chinese medicine and has a long history of application in China [1]. In the last twenty years, many varieties of *G. lucidum* with high fruiting body yield and good quality have been bred at home and abroad and made outstanding contributions to the rapid development of the *G. lucidum* industry [2]. In recent years, many studies have found that *G. lucidum* spore powder had wide activities and effects, such as antitumor effects and immune modulation, due to its various active ingredients [3,4,5,6,7,8]. Moreover, the content of some major ingredients, such as polysaccharides, is much higher than that of *G. lucidum* fruiting bodies [9,10]. *G. lucidum* spore powder products are also edible, convenient and tasty, which makes them more popular with the consumer compared with *G. lucidum* fruiting bodies. So far, almost all varieties are only used for the production of *G. lucidum* fruiting bodies, and the only variety extensively cultivated to obtain *G. lucidum* spore powder is “Longzhi No.1”. However, its poor ability to resist other microbes means it has been gradually abandoned by growers and enterprises, resulting in a dramatic reduction in production. Therefore, more varieties with good resistance to other microorganisms and a high yield of *G. lucidum* spores are urgently needed to replace “Longzhi No.1”.

Currently, many fungal breeding strategies such as crossbreeding [11], protoplast fusion [12], mutagenesis [13] and genetically engineered transformation [14] have been successfully utilized to develop dominant strains in edible and medicinal mushrooms, including *G. lucidum*. Crossbreeding was applied to obtain some strains whose polysaccharide and triterpene content was significantly higher than those of the parental strains [15]. Protoplast fusion of *G. lucidum* with a variety of *G. lucidum*, *G. sense* and other mushroom species, especially those affiliated to different genera of basidiomycetes, has been performed [16], and various fusions were successfully developed. Qi et al. [17]. reported that a new strain of *G. lucidum* with a distinct mutation was acquired after a space flight. Th genetic transformation of *G. lucidum* has also been implemented to express an exogenous glucuronidase gene and the enhanced accumulation of individual ganoderic acids [18]. At present, the fruiting bodies and spores produced from a new *G. lucidum* strain obtained from gene-editing manipulation are prohibited to sold by the Chinese government. Compared with methods such as cross-breeding and protoplast fusion which involve little gene variation, mutagenesis can obtain a large number of mutant strains in a short time, possibly including excellent strains. Artificial mutagenesis using UV irradiation and chemical compounds has been widely conducted to change the basidiospore yield of some mushroom species, including *Pleurotus pulmonarius* [19] and *Pleurotus ostreatus* [20]. In this respect, ultraviolet (UV) ray irradiation may be used to alter the yield of Ganoderma lucidum.

Therefore, considered together, based on a comprehensive comparative analysis of the relevant data from the established germplasm resource information database of *G. lucidum*, the best *G. lucidum* strain, G0109, was selected as the original strain to be induced using UV and regenerated; this process could acquire a novel *G. lucidum* strain with high spore yield and strong resistant ability to undesired microorganisms, which could be used in *G. lucidum* production.

## 2. Materials and Methods

### 2.1. Strains of G. lucidum

*G. lucidum* strain G0109, as the original strain and current main cultivar “Longzhi No.1”, was preserved in the Edible Fungi Sub-Center of the General Microbiology Culture Collection Center in Shanghai, China. They were maintained in potato dextrose agar (PDA; BD, Sparks, NE, USA) slants at 4 °C.

### 2.2. Protoplast Mutagenesis and Identification of Mutagenic Strains

G0109 was transferred to a potato dextrose broth medium (PDB; BD, Sparks, NE, USA) for 2 d of shaking culture at 26 °C followed by 6 d of stationary culture at 26 °C. The protoplast of G0109 was prepared with reference to the modified method [21]. This meant the mycelia were treated at 32 °C for 2.5 h using 2% lywallzyme (Guangdong Institute of Microbiology, Guangzhou, China). A total of 200 μL of protoplast diluted to 10^6^/mL was spread onto the regeneration medium plate, and then the plate was wrapped with aluminum-foil paper and cultured at 26 °C in darkness after it was placed 40 cm from a 20 W UV lamp for 50 s. To avoid photoreactivation, irradiation was implemented in darkness. Tiny regeneration colonies after UV ray treatment were transplanted onto a PDA plate to be cultured at 26 °C in darkness. The regeneration strains with dikaryon were confirmed microscopically by the presence of clamp connections on the mycelia. Somatic incompatibilities between dikaryotic strains and the original G0109strain were detected, and the dikaryotic mutagenic strains were determined by the presence of an antagonistic reaction between them and the original strain.

### 2.3. Mycelia Growth Characteristics of Mutagenic Strains

Mutant strains were firstly activated and then transplanted onto PDA plates with a 5 mm diameter inoculation block to be cultured for 5 days in darkness. The mycelial characteristics of morphology, density and growth rate were observed and measured.

### 2.4. Evaluation of the Resistance Ability to Other Microorganisms and Fruiting of Mutagenic Strains

For investigation of the capacity of resistance to other microorganisms, the yield ratio of good spawns was analyzed. The spawns were prepared in a polypropylene plastic bag (170 × 350 mm) uniformly packed with 1.0 kg of substrate composed of 78% of sawdust, 13% of wheat bran, 7% of corn powder, 1% of sucrose and 1% of gypsum by dry weight, with water content at 65%. The packed bags were sterilized at 121 °C for 2 h. After cooling to room temperature, the top surface of the substrate was inoculated with mycelial blocks growing on the PDA plates and cultured at 24–26 °C in darkness. They were transferred to a cultivation room in which temperature, moisture and illumination intensity were controllable after 30 days of incubation. The spore yield and fruiting bodies yield were also investigated.

### 2.5. Evaluation of the Ability of Mutagenic Strains to Resist Trichoderma

The better acquired mutagenic strains and current main cultivar “Longzhi No.1” in the production regions of *G. lucidum* were cultivated in the same condition. The fruiting body to finished product ratio of their spawn, undesired microorganisms infecting their fruiting bodies, and spore injection in the different growth cycles were investigated to realize the spore yield and resistant ability to undesired microorganisms of the mutagenic strains.

### 2.6. Stability Test of Mutagenic Strains

The better acquired mutagenic strains were sub-cultured for 15 generations; the fruiting experiments of the 1st, 5th, 10th and 15th generation of the mutagenic strains were conducted under the same conditions. Each strain was inoculated in 500–600 bags of solid substrate, as described in the experiment on the evaluation of resistant ability to other microorganisms and the fruiting of mutagenic strains.

### 2.7. Statistical Analysis

All data were shown as mean ± standard deviation (SD). Three experiments were performed with at least three replicates each. The student’s *t*-test was utilized to evaluate the comparisons of two groups, and one-way analysis of variance (ANOVA; SPSS Inc.; Chicago, IL, USA) followed by Tukey’s test was used to assess the differences between multiple groups. Differences were considered to be statistically significant with the *p* value of less than 0.05.

## 3. Results

### 3.1. The Acquisition and Microscopy of Regenerated Single Colony and Identification of Dikaryotic Mutagenic Strains

Regenerated colonies after mutagenesis were transplanted into the center of PDA plates and a sterile cover glass was obliquely inserted at the edge of the colony. When mycelia grew to the glass cover, they were observed under the microscope. Clamp connections with mycelia were named dikaryotic strains (Figure 1A,B). Mutagenic strains were identified using the somatic incompatibilities test between the regenerated dikaryotic strains and original G0109strain. An antagonism effect existed between the dikaryotic mutagenic strains and G0109 (Figure 1C).

### 3.2. Mycelia Growth Characteristics of Mutagenic Strains

Mutant strains were transplanted in PDA plates, using the perforate inoculation method, to be cultured in darkness. The morphology characteristics, density and growth rate of mycelia were investigated (Figure 2A); very slow growth rate and worse growth vigor of mycelia were eliminated. According to the observed morphology characteristics, high density, thickness and rapid growth of mycelia (Figure 2B; Table 1), 55 mutagenic strains were selected for fruiting experiments.

### 3.3. Ability to Resist Other Microorganisms

As shown in Table 2, the yield ratio of good spawns ranged from 57.30% to 100%, of which UV119 was the most resistant to other microorganisms. Those of the mutagenic strains UV48, 105, 136, 160 and 165 reached above 95%. These strains faced the invasion of other microorganisms such as *Penicillium*, *Trichoderma*, *Streptomyces*, *Aspergillus*, and so on, at the stage of their vegetative growth, so the yield ratio of good spawns could largely reflect their capacity to resist other microorganisms.

### 3.4. Bioconversion Efficiency

The spore yield and fruiting body yield were also investigated (Table 3 and Figure 3). If the spawns were only used to produce fruiting bodies, the unit output could achieve 31.20 g per spawn and be higher than that of other mutagenic strains, such as UV80, 100, 117, 156 and 158 (28–29 g per spawn), or the original G0109 strain (28.71 g per spawn). The biological conversion efficiency of the dry substrate of UV119 reached up to 8.91%, which was far higher than the reported 7.49% of “Xianzhi 1” [22].

### 3.5. The Unit Output of Spores

The data from Table 4 indicated that spore unit yield of mutagenic strain UV119 was 19.06 g per spawn and was far higher than that of another mutagenic strain, UV130 (15.76 g per spawn), the original G0109 strain (15.98 g per spawn) and the current main cultivar “Longzhi No.1” (15.81 g per spawn). Hence, whether it was the yield ratio of good spawns and bioconversion efficiency or spore yield per spawn, UV119 exhibited the highest performance of all mutagenic strains.

### 3.6. Ability to Resist Mold of Mutagenic Strain UV119

In field cultivation of *G. lucidum* and related studies, we found that *Trichoderma* seriously inhibited *G. lucidum* vegetative growth, thus decreasing the final yield. As shown in Figure 4, the mutagenic strain UV119 was more resistant to *Trichoderma* invasion than “Longzhi No.1”.

### 3.7. Stability Test of Mutagenic Strain UV119

Stability investigation of the 1st, 5th, 10th and 15th generation mutagenic strain of UV119 was performed. Additionally, the fruiting bodies displayed normal shape and no shape differences existed between these different generations of UV119 (Figure 5). The yield of fruiting bodies and spores was recorded and analyzed. Results showed that there were no significant differences between generations, whether in fruiting body or spore yield (Table 5). Hence, UV119 proved to have better stability and uniformity, and its spore yield was also nearly 20% higher than that of the control cultivar, “Longzhi No.1”. Collectively, there was no doubt that UV119 had good promotion prospects.

### 3.8. The Ability to Resist Undesired Microorganisms and Spore Production Performance of UV119 in Field Cultivation

Some spawn of *G. lucidum* were so seriously contaminated by soil microorganisms that they did not produce fruiting bodies, because no barrier existed between the mycelia and soil while being cultivated; the polypropylene plastic bags wrapped around the surface of the spawn had been removed, covering the spawn in soil (Figure 6). Therefore, the fruiting body production rate could be exploited to preliminarily evaluate the capacity to resist undesired microorganisms. Compared with the current local main cultivar in Zhejiang province, “Longzhi No.1”, the fruiting body production rate of UV119 was up to 99.15% in the first growth cycle (Table 5); it exhibited an excellent ability to resist undesired microorganisms, and it was still up to 88.90% in the second-growth cycle and far better than the 30.80% of “Longzhi No.1” (Figure 7 and Table 6). It could be concluded that mutagenic strain UV119 had a strong ability to resist undesired microorganisms during two whole stages of growth and development. The data of Table 7 showed that under the same cultivation conditions, the spore yield per unit of UV119 increased 26.67% more in the first growth cycle than that of “Longzhi No.1”; in the second-growth cycle, the former was nearly 10 fold higher than the latter, which indicated that UV119 had an extremely significant trait of high spore yield.

## 4. Discussion

Considering that *G. lucidum* is a typical tetrapolar heterothallic fungus that has a double factor incompatibility system composed of incompatibility factors A and B [23], heterokaryons with two different nuclei can produce fruiting bodies and spores. After the induction of *G. lucidum*, protoplasts lack cell walls, and regenerated isolates may contain some monokaryons, which cannot fruit. In the present study, new *G. lucidum* strains were applied to cultivated ones to obtain *G. lucidum* fruiting bodies, especially *G. lucidum* spores. To save time and improve the efficiency of screening better strains, observation of the clamp connection structure which only exists in dikaryotic strains was carried out and followed by a somatic incompatibility test, which was employed to identify whether these regenerated dikaryotic strains were mutants different from an original strain at initial screening. Additionally, 213 of 378 regenerated isolates were eliminated, which greatly reduced screening work.

In edible fungi cultivation, dikaryotic strains with fast substrate colonization and growth capacity are usually preferred because a certain positive correlation exists between mycelia growth rate and fruiting body yield [24]. In this study, in the mutagenesis strains examined, yield did not positively correlate with mycelial growth rate.

A great deal of studies have demonstrated that *G. lucidum* exhibits multiple therapeutic activities, including antitumor, antiviral, immune-modulatory, and antihypertensive activities [25,26,27,28] Additionally, they are becoming more popular and favored among people, with more and more attention being paid to health by society. Therefore, the scale of *G. lucidum* cultivation will be expanded dramatically, and the problem of continuous cropping in *G. lucidum* cultivation will emerge [29]. New *G. lucidum* strains with the ability to resist other microbes’ invasions are the aim of the breeder. In this study, we attempted to develop an evaluation index to select new strains with the ability to resist undesired microorganisms. Here, we firstly reported that the yield ratio of good spawns was constructed to quickly screen the strains. The good spawn ratio of UV119 is up to 100%, and it also exhibits excellent performance in field cultivation.

Our research for the first time achieved a variety of *G. lucidum* with high spore yield and a strong capacity to resist undesired microorganisms, using ultraviolet rays to treat the protoplasts of *G. lucidum*. More cultivation cases demonstrated that its fruiting body and basidiospore yields were, respectively, 8.67% and 19.27% higher than those of the original G0109 strain, which proves that it is feasible and effective to obtain a new strain with a high spore yield using ultraviolet rays to induce *G. lucidum* strains. The present study is consistent with those research findings demonstrating that ultraviolet irradiation can change the basidiospore yield of *Pleurotus eryngii* [30] and *Coprinus cinereus* [31].

Many mutagenic studies on mycelium and the protoplast of *G. lucidum* have been reported, but they were mainly focused on acquiring strains with high biomass, the high content of a certain component or high-activity enzymes for liquid fermentation [32,33], although it has also been reported that the varieties “Hunonglingzhi 1”, “Xianzhi 1” and “Longzhi 2”, utilized in solid cultivation for fruiting bodies, were obtained by mutagenesis of the *G. lucidum* fruiting body (Table 7). However, it has not been reported that strains with the agronomic traits of high spore yield of and an ability to resist undesired microorganisms were bred. Based on the database of *G. germplasm* resource cultivated in China, which was composed of almost all the strains applied in China, strain G0109, with slightly higher spore yield than other congeneric strains, was selected as the original strain. The protoplasts lacking cell walls were more efficiently mutated by ultraviolet ray treatment and many mutation strains were obtained, which would be a good experimental material used to excavate some genes involved in sporulation and basidiospore yield.
foods-12-00465-t007_Table 7Table 7Comparison with recent Ganoderma cultivars.YearsBreeding MethodsCultivar NameCharacteristicsRef.2013UV mutagenesisHunonglingzhi 1The triterpenes content (16.50 mg/g dry weight fruiting bodies) is 15.83% higher than that of the original strain, and also more resistant to other microbes than the original strain.[34]2014UV mutagenesisXianzhi 1Crude polysaccharide content reached up to 24.7 mg/g dry weight of fruiting bodies, which is significantly higher than the original strain[22]2014Screening from some *G. lucidum* strainsLongzhi 11.60 kg basidiospores can be produced per 100 kg of wood substrate[35]2016Domestication of wild *G. lucidum* strainsLongzhi 2The fruiting bodies almost do not produce basidiospores.[36]2016Cross breeding using basidiospore-derived monokaryonsH-23Its polysaccharide content (16.63 mg/g dry weight fruiting bodies) and triterpene content (10.50 mg/g dry weight fruiting bodies)[15]2017Space mutationXianzhi 22.12 kg basidiospores can be produced per 100 kg of wood substrate. However, this variety is never widely applied in *G. lucidum* cultivation.[37]2021Screening from some wild *G. lucidum* strainsXianzhi 3Polysaccharide content of basidiospores is 15.30 mg/g dry weight, 22.40% higher than that of Xianzhi 2.[38]

In conclusion, an ultraviolet ray irradiation method can be utilized to significantly improve the yield of *G. lucidum* spores. The UV119 strain was screened and selected according to some indexes, such as mycelial characteristics, resistance ability to other microorganisms, and spore yield. Its mycelia are tiled and dense. Additionally, its mycelial layer is thick, and the good spawn ratio is up to 100% under laboratory conditions. Its fruiting body and basidiospore yields are, respectively, 8.67% and 19.27% higher than those of the original G0109 strain; and the basidiospore yield in particular is 20.56% higher than that of the current main cultivar “Longzhi No.1”. It is worth mentioning that in large-scale field cultivation UV119 shows more excellent performances than the latter, and also exceeds the variety “Xianzhi 2”, bred using space mutation. Therefore, it will be favored and widely applied in the near future.

## Figures and Tables

**Figure 1 foods-12-00465-f001:**
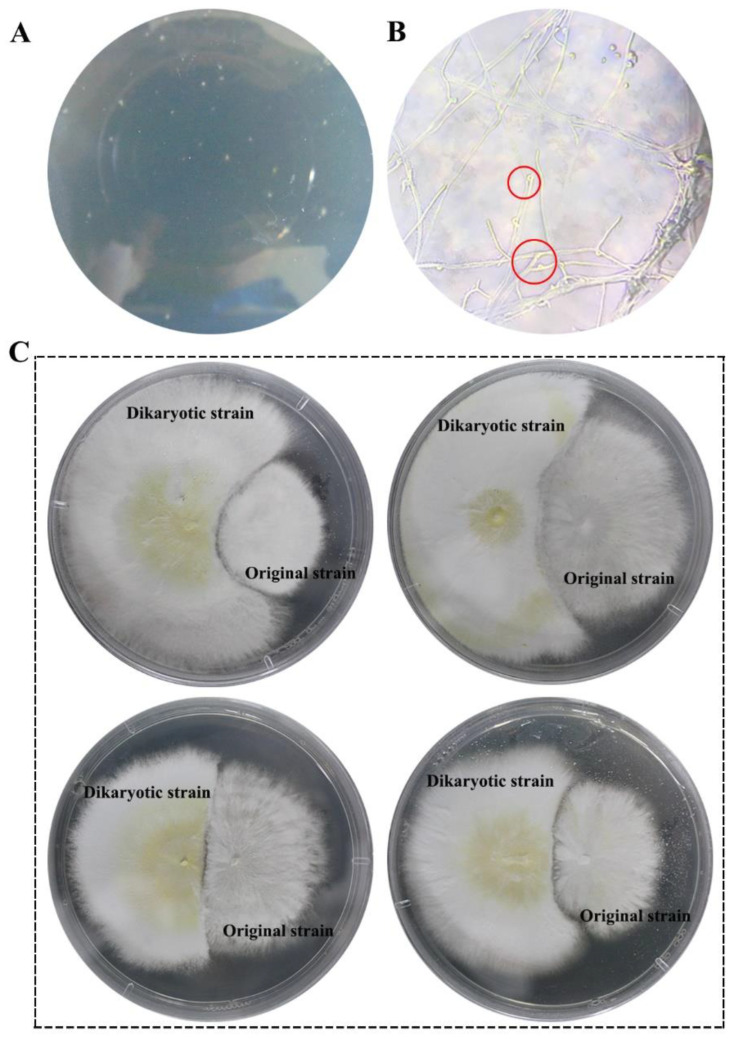
The acquisition and microscopy of a regenerated single colony and identification of dikaryotic mutagenic strains. (**A**): Regeneration colonies after protoplast mutagenesis of G0109; (**B**): clamp connection of regenerated dikaryotic strains; (**C**): Somatic incompatibility reaction between dikaryotic strains and original strain.

**Figure 2 foods-12-00465-f002:**
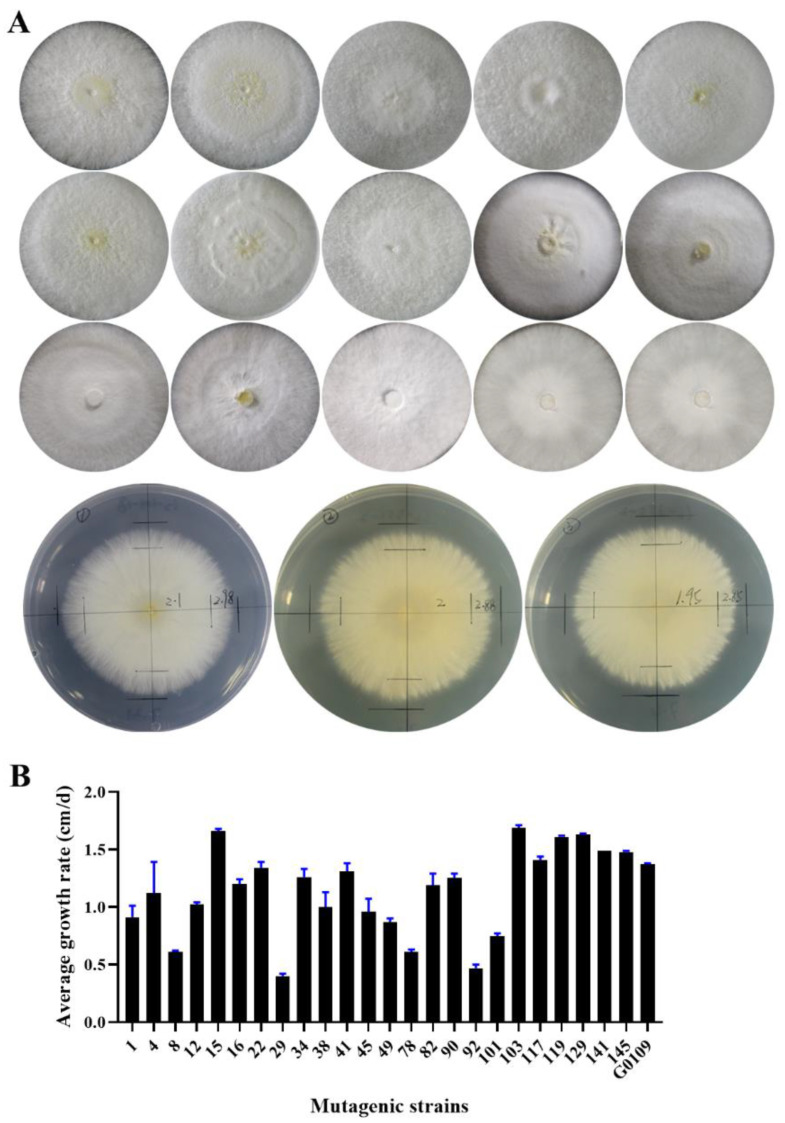
Mycelia growth characteristics of mutagenic strains. (**A**): Mycelia morphology and growth rate of some mutagenic strains; (**B**): Mycelia growth rate of some mutagenic strains.

**Figure 3 foods-12-00465-f003:**
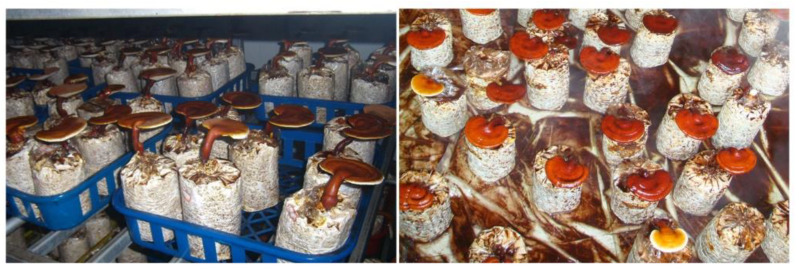
Fruiting of the 1st, 5th, 10th and 15th generations of the mutagenic strain UV119.

**Figure 4 foods-12-00465-f004:**
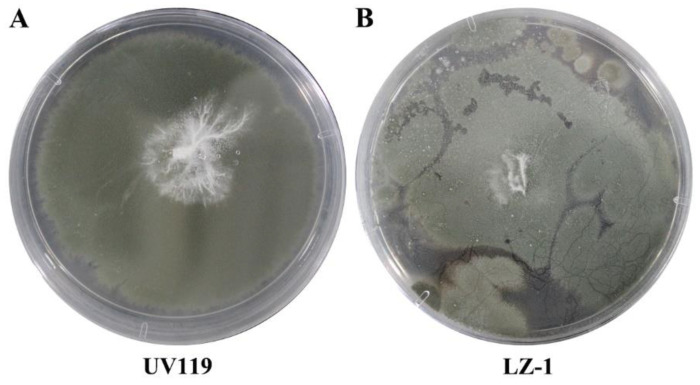
Growth of mutagenesis on PDA plate dominated by *Trichoderma*. LZ-1: Longzhi No.1. (**A**): UV119 grew on PDA plate with *Trichoderma*; (**B**): LZ-1 grew on PDA plate with *Trichoderma*.

**Figure 5 foods-12-00465-f005:**
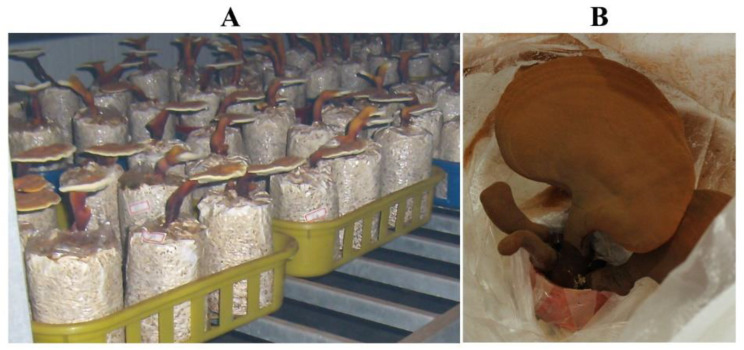
Fruiting of the 1st, 5th, 10th and 15th generations of the mutagenic strain UV119. (**A**): Fruiting situation of different generations of UV119; (**B**): collection of spores powder.

**Figure 6 foods-12-00465-f006:**
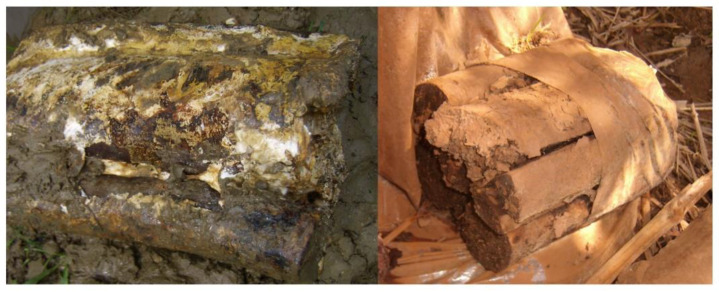
Some spawn of *G. lucidum* were contaminated by soil microorganisms.

**Figure 7 foods-12-00465-f007:**
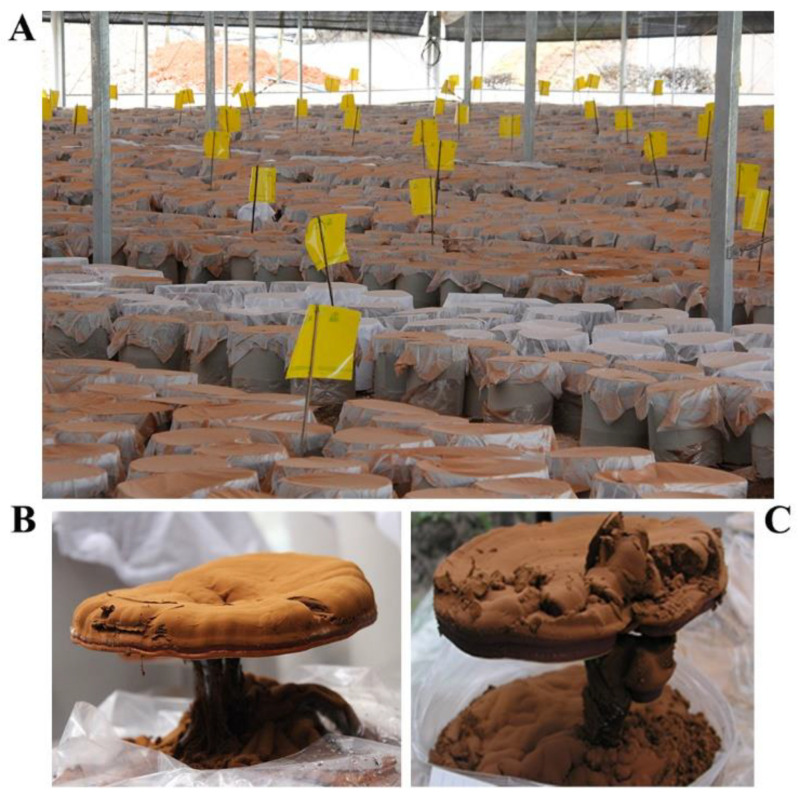
Production of spores during the strain cultivation. (**A**): Spore collection; (**B**): LZ-1; (**C**): mutagenic strain UV119.

**Table 1 foods-12-00465-t001:** Mycelia characters of some mutagenic strains.

No.	Morphology	Color	Density	Notes
1	Tile Type	white	+++	
4	Villiform	white	+++	
8	Tile Type	white	+++	thin
12	Tile Type	white	+++	thin
15	Tile Type	white	++	thin
16	Villiform	white	++	
22	Tile Type	white	++	
28	Villiform	white	+++	
29	Transitional Type	white	++	thin
34	Tile Type	white	+++	
41	Tile Type	white	+++	thin
43	Transitional Type	white	+++	
49	Tile Type	white	++	thick
52	Transitional Type	white	+	
55	Tile Type	white	++	thick
62	Villiform	white	+++	thin
75	Tile Type	white	+++	thick
78	Transitional Type	white	+++	thin
82	Transitional Type	white	+++	
90	Villiform	white	+++	
92	Tile Type	white	+++	thin
101	Tile Type	white	++	thin
103	Villiform	white	++	
117	Tile Type	white	++	thick
119	Tile Type	white	+++	thick
129	Villiform	white	+++	
141	Transitional Type	white	++	thin
145	Tile Type	white	+++	
146	Tile Type	white	+++	thin
155	Transitional Type	white	+++	
158	Tile Type	white	++	thick
161	Transitional Type	white	+	
165	Tile Type	white	++	thick

+: mycelia is very sparse; ++: mycelia is spares; +++: mycelia is bushy; ++++: mycelia is very bushy.

**Table 2 foods-12-00465-t002:** Yield ratio of good spawns (YRGS) of spawns originating from mutagenic strains.

No.	YRGS (%)	No.	YRGS (%)
1	81.33 ± 1.20	68	78.33 ± 0.88
3	70.34 ± 1.45	75	71.53 ± 0.98
5	65.00 ± 1.53	76	68.30 ± 0.85
6	72.46 ± 1.44	79	57.30 ± 1.20
9	88.62 ± 1.76	80	83.33 ± 1.12
10	87.67 ± 1.86	88	90.67 ± 1.21
12	93.00 ± 1.53	89	78.50 ± 1.30
13	82.00 ± 1.54	93	68.70 ± 0.89
14	70.00 ± 1.53	100	78.46 ± 1.02
17	78.80 ± 1.67	105	98.33 ± 0.88
18	86.34 ± 0.88	117	93.68 ± 0.96
19	91.67 ± 0.65	119	100.00 ± 0.00
21	62.54 ± 0.48	130	64.48 ± 1.21
23	73.78 ± 0.88	136	96.70 ± 0.78
25	94.70 ± 0.67	137	91.36 ± 1.03
26	70.00 ± 0.81	140	82.33 ± 1.45
30	86.50 ± 0.86	142	87.76 ± 0.85
34	91.70 ± 1.20	143	91.67 ± 1.03
39	72.80 ± 0.98	145	94.30 ± 0.90
40	81.77 ± 1.09	146	87.34 ± 1.20
41	89.33 ± 0.34	148	75.67 ± 1.76
43	60.38 ± 0.97	154	67.33 ± 0.91
47	78.70 ± 0.75	156	90.30 ± 1.46
48	95.67 ± 0.43	158	86.40 ± 1.04
50	90.34 ± 0.78	160	96.40 ± 0.89
53	87.34 ± 0.67	162	87.66 ± 0.92
55	81.30 ± 0.71	165	98.70 ± 0.85
62	91.00 ± 1.00	G0109	97.70 ± 1.00

**Table 3 foods-12-00465-t003:** The unit output (gram per spawn) of mutagenic strains.

No.	Yield per Unit	No.	Yield per Unit
1	25.92 ± 1.33	68	24.67 ± 0.99
3	26.19 ± 0.16	75	24.44 ± 1.55
5	24.66 ± 0.61	76	25.99 ± 1.49
6	24.43 ± 1.10	79	25.02 ± 1.13
9	25.15 ± 1.67	80	28.62 ± 1.37
10	24.41 ± 1.02	88	25.29 ± 0.81
12	24.24 ± 0.75	89	24.42 ± 1.13
13	26.95 ± 1.20	93	22.44 ± 0.73
14	25.11 ± 1.44	100	28.92 ± 0.66
17	25.21 ± 0.76	105	24.27 ± 1.33
18	24.65 ± 1.80	117	28.45 ± 1.43
19	26.17 ± 1.15	119	31.20 ± 0.93
21	25.65 ± 1.00	130	27.03 ± 0.98
23	22.67 ± 0.63	136	25.50 ± 1.66
25	25.43 ± 1.11	137	24.22 ± 1.38
26	24.08 ± 1.09	140	27.18 ± 0.95
30	26.56 ± 1.45	142	26.43 ± 0.19
34	24.78 ± 3.19	143	22.99 ± 2.05
39	23.61 ± 0.75	145	28.83 ± 0.87
40	24.75 ± 1.09	146	23.76 ± 0.99
41	24.80 ± 0.73	148	26.84 ± 4.73
43	27.01 ± 1.77	154	27.20 ± 1.86
47	25.56 ± 0.86	156	28.42 ± 0.94
48	27.52 ± 1.19	158	29.32 ± 1.03
50	22.77 ± 1.99	160	26.41 ± 3.81
53	27.26 ± 1.56	162	23.33 ± 1.50
55	26.46 ± 1.39	165	27.33 ± 0.98
62	25.06 ± 0.87	G0109	28.71 ± 0.98

**Table 4 foods-12-00465-t004:** Spore unit output (gram per spawn) of some mutagenic strains.

No.	Spores Unit Output	No.	Spores Unit Output
1	5.11 ± 0.10	68	4.62 ± 0.25
3	8.44 ± 0.20	76	3.18 ± 0.22
5	6.49 ± 0.11	79	6.53 ± 0.14
6	5.73 ± 0.09	80	5.69 ± 0.21
9	6.63 ± 0.30	88	12.69 ± 0.20
10	8.41 ± 0.19	89	10.01 ± 0.15
12	7.31 ± 0.10	93	7.68 ± 0.33
13	10.77 ± 0.19	100	4.13 ± 0.21
14	3.67 ± 0.15	105	5.59 ± 0.27
17	4.15 ± 0.15	117	2.71 ± 0.19
18	5.71 ± 0.27	119	19.06 ± 0.15
19	6.68 ± 0.23	130	15.76 ± 0.18
21	8.18 ± 0.22	136	3.37 ± 0.19
23	2.75 ± 0.20	137	8.81 ± 0.16
26	2.28 ± 0.08	140	3.52 ± 0.15
30	9.06 ± 0.15	143	5.51 ± 0.20
34	6.70 ± 0.23	145	3.22 ± 0.17
39	10.23 ± 0.13	146	9.24 ± 0.15
40	7.69 ± 0.11	148	4.63 ± 0.21
41	4.43 ± 0.15	154	3.62 ± 0.18
43	6.52 ± 0.24	156	6.62 ± 0.20
47	3.67 ± 0.24	158	5.82 ± 0.18
48	5.93 ± 0.13	160	3.40 ± 0.12
50	4.43 ± 0.14	162	2.75 ± 0.19
53	7.29 ± 0.10	165	4.70 ± 0.20
55	12.77 ± 0.12	G0109	15.98 ± 0.27
62	7.07 ± 0.14	LZ-1	15.81 ± 0.20

LZ-1: Longzhi No.1 was used to produce *G. lucidum* spores. The spore unit output of the strains UV25, 62 and 75 could not be counted and analyzed due to collection confusion.

**Table 5 foods-12-00465-t005:** Spore and fruiting body unit output of different generation subcultures of UV119.

Strains	Fruiting Bodies Yield (g/Spawn)	Spores Yield (g/Spawn)
UV119	31.73 ± 0.80	19.32 ± 0.35 **
UV119-5	31.28 ± 0.51	18.94 ± 0.35 **
UV119-10	32.07 ± 0.61	18.87 ± 0.36 **
UV119-15	31.37 ± 0.99	19.12 ± 0.32 **
LZ-1	#/	15.95 ± 0.48

** *p* < 0.01, compared with the LZ-1, #: Fruiting bodies Yield of LZ-1 was not presented here due to the comparison of spores powder yield with UV119.

**Table 6 foods-12-00465-t006:** Fruiting rate and spores yiled per unit of the two varieties in the two growth cycles.

Site	Strain	Log Cultivation (Bag) ^a^	First Growth Cycle		Second Growth Cycle	
			FBPR (%) ^b^	SUY	FBPR (%)	SUY
Longquan, Zhejiang	UV119	6500	99.15 ± 0.83 ***	0.19 ± 0.01 *	88.90 ± 1.41 ***	0.10 ± 0.01 ***
	LZ-1	6600	90.16 ± 0.91	0.15 ± 0.02	30.80 ± 0.95	0.01 ± 0.00

a: indicated 10 kg wet weight of spawn per bag; b: kg per bag. * *p* < 0.05, *** *p* < 0.001, compared with the LZ-1. FBPR: Fruiting body production rate; SUY: Spores unit yield.

## Data Availability

The data used and analyzed during the current study are available from the corresponding author on academic request (H.Z.). The data are not publicly available to preserve the privacy of the data.

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
