# Peer review of "A Novel Strain Breeding of Ganoderma lucidum UV119 (Agaricomycetes) with High Spores Yield and Strong Resistant Ability to Other Microbes’ Invasions"

_foods, 2023, doi:10.3390/foods12030465_

Round 1

Reviewer 1 Report

1. UV irradiation: Is the selected exposure time of 50 secs considered the best for the mutagenic process? Preliminary studies? Previous studies? Be specific.

2. It would be best to include the recyclability efficacy of the prepared growth solid substrate block (with that mutated gen) as compared to the control. Please include the dataset if possible for better understanding.

3. What is the specification of the parameters involved during the propagation process (especially in field cultivation)? e.g. temperature/humidity/area/harvesting cycles/vapour pressure. I did not come across to see any of these parameters mentioned elsewhere. 

 4. I don't see any conclusive evidence to confirm that the irradiation process would possibly alter the gen at the molecular level of the used spores. It was just a physical effect in terms of its yield and none of the confirmation was presented in the main text. Changes to the strain need to be carried out via physico-chemical analysis or perhaps details molecular level profiles to confirm it definitely.   

Author Response

Q1. UV irradiation: Is the selected exposure time of 50 secs considered the best for the mutagenic process? Preliminary studies? Previous studies? Be specific.

Response: Thank you very much for your good comments. In this study, the selection experiment of UV irradiation time was carried out according to 0, 5, 10, 15, 20, 25, 30, 35, 40, 45, 50, 55, and 60 seconds (supplementary figure 1). When the irradiation time was in 50 seconds, the fatality rate was about 80.30%. According to the previous studies [1-5], there would be more positive mutations as the lethality rate reached 75%-80%. Therefore, the optimal time of UV irradiation was selected as 50 seconds.

Supplementary figure 1. The morphology of G. lucidum mycelia at different times of UV irradiation treatment

[1]   Sen ZQ, Xing B, Ding ZD, et al. Mutagenesis breeding of D-lactonohydrolase high-producing bacteria and optimization of fermentation conditions. Food and Fermentation Industries. 2022. doi.org/10.13995/j.cnki.11-1802/ts.031194.

[2]   Liu YQ, Luo LY, Liu HR, et al. Breeding of a High Temperature Resistant Strain of Ganoderma lucidum. Edible fungi of China, 2021, 40(8):13-17.

[3]   Dong YW, Miao JZ, Cao ZH, et al. UV Mutagenization of protoplast for screening and breeding of high-yield Organog emanium-producing Strains of Ganoderma lucidum. Food Science, 2009, 30(15):188-192.

[4]   Yan F, Zhang HS, Li RB, et al. Screening of high yield polysaccharide mutants by Protoplast UV mutagenesis. China brewing, 2008, (22):25-29.

[5]   Zhang H, Li CB, Chen MB, et al. Mutagenesis breeding new strain of Jisongrong mushroom (Agaricus blazei) by UV. Journal of Microbiology, 2004, (06):56-57.

Q2. It would be best to include the recyclability efficacy of the prepared growth solid substrate block (with that mutated gen) as compared to the control. Please include the dataset if possible for better understanding.

Response: Thanks for your suggestion. When G. lucidum fruiting bodies developed mature, many basidiospores started ejecting out of the pileus and were collected. At the tubes (lower layer of pileus) was the sporogenic structure (Supplementary figure 2). The two tubes at the same developmental stages were observed using an electronic microscope. It could be seen that the basidiospores quantity from WT was less than that of MT, which was consistent with the practical situations. Moreover, a study by Okuda et al. [6] showed that an msh4 homolog, stpp1, from Pleurotus pulmonarius, was related to the spore’s production, its 193 bp length region deleted in the spore-less mutant strain. The spore-less trait was temperature independent and had no linkage to unfavorable traits affecting fruiting, spore germination, or mycelial growth. Therefore, the msh4 homolog from G. lucidum was cloned and sequenced. Two little fragments (22 bp, 20 bp) deletion was found through an alignment using Mega 7.0 software.

Supplementary figure 2. Inner structure of the tube (sporogenic structure) of G. lucidum fruiting bodies under electronic microscope (at the same developmental stage)

Note: Left: wild type G0109;Right: mutant type UV119; Up: 100 μm; down: 30 μm. Arrows represented G. lucidum basidiospores

[6]     Okuda Y, Murakami S, Honda Y, et al. An MSH4 Homolog, stpp1, from Pleurotus pulmonarius is a "Silver Bullet" for resolving problems caused by spores in cultivated mushrooms. Applied & Environmental Microbiology, 2013, 79(15): 4520-4527.

Sequences of msh4 of G. lucidum were as follows.

begin characters;

       dimensions nchar=3372;

       format missing=? gap=- matchchar=. datatype=nucleotide;

       matrix

MT(UV119):

ATGCATCTCCACTATCCTTCAATCGTCCTTGTTCCTGACACGTTCATGTCCCTGTCTGATGTTTCGCTCCCATCTGGTGCGAAGACCCCGCAGACGACAACACTTCTCGTTCAATGCATCATGGACGAGTTCGACGGTGTACCCATCGAGCCAGTCATGCGCAAATATTGGAGCGATAACGCAGGTGAGTACTCGGGTCTCTCGCTTGCGTGCACAGCACTGATAAATACATTAGGGCTAGACTTCATCAACCAGCTCATGGTCGACAACGACGAACGTGCTGCCACCCTCGTCGCCGTGTCCAATAAGTGCATATATGCACCTAAACTCTAGCATCTCTTGCGTGCTCAACATTGCCACAGATACTATGCACTCTCTGCCGCCAGCGCGCTCTTCAAGCATGCTGAACTCAAGCTCAACGTGCGCTTTGCTGCTTCGTCTCTACTTATCCGATACACCCAGGTGGAAGGTACAATGATGATTGACTCTGACACCGCTCGCAATCTCGAGCTGGTTGGCAACATGTCAGTTCGGAAGAGTGCGCATTCTCTTTTCGCGTACGTTCATATGTCTTCATTTAGCTCAAGCTTGGTGGTGACTCAGATATATGCGCAGTTTGTTGAACCATACGTATACGGCCATGGGCGCAAGACTGTTGAGGGTCAACATTTTAGCTCCTATCACTGGTCAGTCTCGTGCAACAGCGTTCGTTGAACCCAAGAGTTAACATATGCATATGCTGCCAGTGAAATCTGCACTGGAAGCACGACTAGATGTAGTCGAAGGTACAATCATCTTGCTGTCAGTGTATAAGTATCTAATCCAAATCTAATCACGTCTCGACTAGAGTTCATCCAAACTGAAGACCGTTTCACGGACGTCAAGGACGCTCTCAAATCACTTAACAAGCTCGATTTCGACAAGCTTATTTGTTCTGTGCGTCTCTTCAAGGCCACCGCTTCAAGGACACTGTGCTGGGAGATATTCATGTCGAATGCAGCTCGCATCCTCCGAAGTTCGAGAAGTTAGCACCTCAAAAATAGCATCGGCGAGAGTCGCACAAATGCTCAACCTCAGGAACATTGTCCAAAGTCTCCCCCGGCTTGCCAAGGCTCTTGAGGGAAGTCGGTCGCAGCTGTTACAAATCATCGCAGAGGTTCGCCTTAGTCTGTATCTCGAGTACCACGGCTGTGCTTAGTGCCAAGGACTATCCTCAGATGATATCAGACGATCGTCTGGAGAAGATCGAGCGGTTGGTATGCGACCGGTTGAATGAGGAGACAACTCCCGCAAAGGTAATCGTCATGCAATGCCATGATTATCGAGACTCTTAACATCCATATCAAGGGAGGGTTGAATGCGGTGAACGCTAGAGTCTACGCAGTTAAGGTAATGTCTGCATTGTCTGCCGAGGACTTGATTGAACGAATCTTCCATCAGGCTAATTTCAATCGCCTGCTCGACGTTGCTCGTGAAACCTATAAGGAGAACGTCGGCGACATCTATGCTCTGCGCAATCGTCTTGCCGAAGAACACAACCTCCCTTTCACTCTAGTGTACCGCGATTCTGATGCGGGCTTTGTCTTCTATATCAAGAAGACTGACCTTGACGAAGCTGGCGGCGAGCTTCCTCGCGGCTTCATCGACGCGATGGCACAGAAAGGAAGATGGGTGTTCTCGAGTATCGAGCTGGTGAGGAGGATATTCCCTATGTACATCGGCAAGAAGAGCCTGAACATCAACAGAAAAAGAGAAACGCACGCATGAAGGATGCGCTGGACGAGTCCTTGATCCTTAGTGACAAGCGAGTACATCGGATTTCTAGTGGCTTACATCGGTCTGAAGTGCGTCTCAAGGATCATTCAGGACCTCACAGACGAGATAGTCGTCGATATGGGTGCTCTGTACAAGGCATCCGAAGCAGTAGCGCTCCTTGATCTCTTATGGT----------------------CATTTGCGCATGCCTCAATCCGTGAGCCCTTCGTTTGTTGTGTTTTCACCCCGCGCTTACAAATTTTAGTGCGAAACTATGGTTCGTCCCTGAGTACTCCCCCGCGATGCCTTCCCGAAGGCTCATGTAGCTGTGCTGTGCGACAGTGCGCCCGGAGTTCACTGGGACTCTTGCAATCAAAGGTGGCCGCCATCCGGTTCTGGAATGTGTACAGTCTGCCGGCGCGACTATCCCCAACGACGTTTATTGTTGCGAGGCGTCTTCATTCCAGATTGTTCAAGGTCCAAAGTACGTGTCCCGTACTGGTTGGTTTTCCTACGCCACTGGGATATTGAGCTGGTTTGCCCCAGCATGTCGGGCAAAAGCACGTATCTACGACAGATTGCCCTGATAACAGTGATGGCAATGTGTGGTTGTTTCATTCCCGCAGAGTATGGATCATTCAGGTACACTCTAATGTGCTAACCTCTGTGGTTTGTTTTGAACAACGCGCCGACATATCCAGGATTCATGATCATCTGCTTACCCGACTTTCGAATGATGACGATCTGGAGAAGAACCTGAGCACGTTCGCGAATGAAATGGCCTCAACTGCCATGATTCTAGGTACATTTCAACTATGTCCCGTGCGCACCTCTCTCATTCTGACTAGGTCTGGCCACTGAGAACTCCCTGGTACTCATCGATGAAGTGGGTCGGGGTACCTCGGCGAGAGAAGGCGTAGCTATCTCACATGCCATTGCGGAGGAGCTCAT--------------------GGTCCCTTCATCTTATAAGCCCACTCTGTCTAACCATTCAAACCAGTCCTTTGTCTTCTTTGCAACGCAAGGGTCCACTACAATTCTAACCCGTCAGCCTTTCACTGATGGTCAGATCTCAGACATTTCAACAAGCTCACCACTACACTCTCCCGACAGCCATCTGTCGTCAAGTAAACTTTTACCGGTGGCTTCCTCCAAGGGTGGGTACTGACGCCGTCCCCTACAGTTTACATCTATCCTTCCAGGTACTCTTTTCCTATAACCTGCCTGTTTTGTGAACTACGTGTTCAACAAATTACAGAAATCACTGCCGACAGCCTCTAAAGTTGGCATCACTTTCCACTACAAGTAAGAAGCTGTCTCTTTATCCGCTCTAAGCCTCTCCTCATCAGCCCTTAGGATTATGGATGGCGCTCCGGAAACTCAAGACCATTATGGTATGATCGTTACCTATTGAACGCTCACATACTGCCAAACAGTCGTCTAGGCTTAGACCTTGCTCGTTTGGCGGACCTCCCAGAGTCCGTTGTATCCGAGGCTCGACGTGTCGCTGAGTACCTCGCGAAACAAGAGGAGCGGGACCAGCAGCAGAGTAAGACGAGCAAGATTGCGCTCCGCCGCAAA

WT(G0109):

ATGCATCTCCACTATCCTTCAATCGTCCTTGTTCCTGACACGTTCATGTCCCTGTCTGATGTTTCGCTCCCATCTGGTGCGAAGACCCCGCAGACGACAACACTTCTCGTTCAATGCATCATGGACGAGTTCGACGGTGTACCCATCGAGCCAGTCATGCGCAAATATTGGAGCGATAACGCAGGTGAGTACTCGGGTCTCTCGCTTGCGTGCACAGCACTGATAAATACATTAGGGCTAGACTTCATCAACCAGCTCATGGTCGACAACGACGAACGTGCTGCCACCCTCGTCGCCGTGTCCAATAAGTGCATATATGCACCTAAACTCTAGCATCTCTTGCGTGCTCAACATTGCCACAGATACTATGCACTCTCTGCCGCCAGCGCGCTCTTCAAGCATGCTGAACTCAAGCTCAACGTGCGCTTTGCTGCTTCGTCTCTACTTATCCGATACACCCAGGTGGAAGGTACAATGATGATTGGCTCTGACACCGCTCGCAATCTCGAGCTGGTTGGCAACATGTCAGTTCGGAAGAGTGCGCATTCTCTTTTCGCGTACGTTCATATGTCTTCATTTAGCTCAAGCTTGGTGGTGACTCAGATATATGCGCAGTTTGTTGAACCATACGTATACGGCCATGGGCGCAAGACTGTTGAGGGCCAACATTTTAGCTCCTATCACTGGTCAGTCTCGTGCAACAGCGTTCGTTGAACCCAAGAGTTAACATATGCATATGCTGCCAGTGAAATCTGCACTGGAAGCACGACTAGATGTAGTCGAAGGTACAATCATCTTGCTGTCAGTGTATAAGTATCTAATCCAAATCTAATCACGTCTCGACTAGAGTTCATCCAAACTGAAGACCGTTTCACGGACGTCAAGGACGCTCTCAAATCACTTAACAAGCTCGATTTCGACAAGCTTATTTGTTCTGTGCGTCTCTTCAAGGCCACCGCTTCAAGGACACTGTGCTGGGAGATATTCATGTCGAATGCAGCTCGCATCCTCCGAAGTTCGAGAAGTTAGCACCTCAAAAATAGCATCGGCGAGAGTCGCACAAATGCTCAACCTCAGGAACATTGTCCAAAGTCTCCCCCGGCTTGCCAAGGCTCTTGAGGGAAGTCGGTCGCAGCTGTTACAAATCATCGCAGAGGTTCGCCTTAGTCTGTATCTCGAGTACCACGGCTGTGCTTAGTGCCAAGGACTATCCTCAGATGATATCAGACGATCGTCTGGAGAAGATCGAGCGGTTGGTATGCGACCGGTTGAATGAGGAGACAACTCCCGCAAAGGTAATCGTCATGCAATGCCATGATTATCGAGACTCTTAACATCCATATCAAGGGAGGGTTGAATGCGGTGAACGCTAGAGTCTACGCAGTTAAGGTAATGTCTGCATTGTCTGCCGAGGACTTGATTGAACGAATCTTCCATCAGGCTAATTTCAATCGCCTGCTCGACGTTGCTCGTGAAACCTATAAGGAGAACGTCGGCGACATCTATGCTCTGCGCAATCGTCTTGCCGAAGAACACAACCTCCCTTTCACTCTAGTGTACCGCGATTCTGATGCGGGCTTTGTCTTCTATATCAAGAAGACTGACCTTGACGAAGCTGGCGGCGAGCTTCCTCGCGGCTTCATCGACGCGATGGCACAGAGAGGAAGATGGGTGTTCTCGAGTATCGAGCTGGTGAGGAGGATATTCCCTATGTACATCGGCAAGAAGAGCCTGAACATCAACAGAAAAAGAGAAACGCACGCATGAAGGATGCGCTGGACGAGTCCTTGATCCTTAGTGACAAGCGAGTACATCGGATTTCTAGTGGCTTTCATCGGTCTGAAGTGCGTCTCAAGGATCATTCAGGACCTCACAGACGAGATAGTCGTCGATATGGGTGCTCTGTACAAGGCATCCGAAGCAGTAGCGCTCCTTGATCTCTTATGGTGCGCTCCTTGATCTCTTATGGTCATTTGCGCATGCCTCAATCCGTGAGCCCTTCGTTTGTTGTGTTTTCACCCCGCGCTTACAAATTTTAGT

GCGAAACTATGGTTCGTCCCTGAGTACTCCCCCGCGATGCCTTCCCGAAGGCTCATGTAGCTGTGCTGTGCGACAGTGCGCCCGGAGTTCACTGGGGCTCTTGCAATCAAAGGTGGCCGCCATCCGGTTCTGGAATGTGTACAGTCTGCCGGCGCGACTATCCCCAACGACGTTTATTGTTGCGAGGCGTCTTCATTCCAGATTGTTCAAGGTCCAAAGTACGTGTCCCGTACTGGTTGGTTTTCCTACGCCACTGGGATATTGAGCTGGTTTGCCCCAGCATGTCGGGCAAAAGCACGTATCTACGACAGATTGCCCTGATAACAGTGATGGCAATGTGTGGTTGTTTCATTCCCGCAGAGTATGGATCATTCAGGTACACTCTAATGTGCTAACCTCTGTGGTTTGTTTTGAACAACGCGCCGACATATCCAGGATTCATGATCATCTGCTTACCCGACTTTCGAATGATGACGATCTGGAGAAGAACCTGAGCACGTTCGCGAATGAAATGGCCTCAACTGCCATGATTCTAGGTACATTTCAACTATGTCCCGTGCGCACCTCTCTCATTCTGACTAGGTCTGGCCACTGAGAACTCCCTGGTACTCATCGATGAAGTGGGTCGGGGTACCTCGGCGAGAGAAGGCGTAGCTATCTCACATGCCATTGCGGAGGAGCTCATCCGACTGAAAGTACGGACATGGTCCCTTCATCTTATAAGTCCACTCTGTCTAACCATTCAAACCAGTCCTTTGTCTTCTTTGCAACGCAAGGGTCCACTACAATTCTAACCCGTCAGCCTTTCACTGATGGTCAGATCTCAGACATTTCAACAAGCTCACCACTACACTCTCCCGACAGCCATCTGTCGTCAAGTAAACTTTTACCGGTGGCTTCCTCCAAGGGTGGGTACTGACGCCGTCCCCTACAGTTTACATCTATCCTTCCAGGTACTCTTTTCCTATAACCTGCCTGTTTTGTGAACTACGTGTTCAACAAATTACAGAAATCACTGCCGACAGCCTCTAAAGTTGGCATCACTTTCCACTACAAGTAAGAAGCTGTCTCTTTATCCGCTCTAAGCCTCTCCTCATCAGCCCTTAGGATTATGGATGGCGCTCCGGAAACTCAAGACCATTATGGTATGATCGTTACCTATTGAACGCTCACATACTGCCAAACAGTCGTCTAGGCTTAGACCTTGCTCGTTTGGCGGACCTCCCAGAGTCCGTTGTATCCGAGGCTCGACGTGTCGCTGAGTACCTCGCGAAACAAGAGGAGCGGGACCAGCAGCAGAGTAAGACGAGCAAGATTGCGCTCCGCCGCAAA

Q3. What is the specification of the parameters involved during the propagation process (especially in field cultivation)? e.g. temperature/humidity/area/harvesting cycles/vapour pressure. I did not come across to see any of these parameters mentioned elsewhere.

Response: Thanks for your comments. During the development of the fruiting bodies in cultivation room, the temperature was controlled at 26-28˚C and the humidity was regulated at 85%-95% using the humidifier, and adjusted to 80% or so when spores was collected for 30 d using non-woven cylinders lined with food preservation bags.

As to field cultivation of G. lucidum using wood as substrates, the wood about 10 kg weight, packed with polypropylene bags should be sterilized for 36-40 hours at 98-100˚C. Inoculation of wood bags was performed according to the ratio of about 500 g mycelia including sawdust per wood bag. And then the wood bags were put in a culture room with a temperature of 20-24˚C for 70-80 d. During the mycelial growth under dark condition, CO2 concentration was kept below 2000 ppm, and humidity controlled at 70%. The physiological maturity sign of wood bag was that some reddish-brown spots or a small number of primordia existed on the surface of wood.

The matured wood bags were put in the filed for 2 weeks and then the polypropylene bags were removed from the surface of the wood. 3000 wood bags were uniformly set in the field of 667 m2. During the development of the fruiting bodies, the temperature was controlled at 24-28˚C and the humidity was regulated at 85%-95% by drop irrigation, and adjusted to 75%-85% when spores was collected for 2-2.5 months using hard paper cylinders lined with food preservation bags.

Q4. I don't see any conclusive evidence to confirm that the irradiation process would possibly alter the gen at the molecular level of the used spores. It was just a physical effect in terms of its yield and none of the confirmation was presented in the main text. Changes to the strain need to be carried out via physico-chemical analysis or perhaps details molecular level profiles to confirm it definitely.

Response: The answer to this question has been finished when I made a response to the second question.

Reviewer 2 Report

This manuscript was well written with an interesting method for Ganoderma strain improvement and development spore production. The experiments were carried out properly however, just simple points below should be asked to the authors.

1.       Why the authors choose 15 generations for stability test? Do you have any reference for stability test?

2.       Should the Figure caption be under the Figure? Please check the format again

3.       There are some places you forgot to make the scientific names italicized. Please check them again

Author Response

Dear Reviewer,

We would like to thank you for your efforts concerning our manuscript entitled “A novel strain breeding of Ganoderma lucidum UV119 (Agaricomycetes) with high spores yield and strong resistant ability to other microbes’ invasions” (foods-2070359). Those comments are all valuable and very helpful for revising and improving our paper, as well as the important guiding significance to our researches. We have studied comments carefully and have made correction which we hope meet with approval. Revised portion are marked in red in the paper. The main corrections in the paper and the responses to the reviewer’ comments are as follows:

Reviewer(s)' Comments to Author:

Reviewer: 2

This manuscript was well written with an interesting method for Ganoderma strain improvement and development spore production. The experiments were carried out properly however, just simple points below should be asked to the authors.

Q1. Why the authors choose 15 generations for stability test? Do you have any reference for stability test?

Response: Generally, it is thought that new strains produced by UV mutagenesis may have reverted mutations [1]. In our study, it is also true that some mutant strains showed reversion of mutations. So experiments on stability should be performed. Subculture is usually used to verify the stability between different generations of strains. Li et al. [1] found that the two mutant strains induced by UV mutagenesis were excellent strains with stable properties of high production and high polysaccharide content after the test of subculture for 10 generations on PDA slants and successive tests of shaking flask and pilot scale. Han et al. [2] demonstrated that Volvariella volvacea VH strain mutated by a combination of UV irradiation, 60Co-γ rays and DES was genetically stable and higher by over 200% than the original strain V23 in biological efficiency after the cultivation test of Gen. 1, 5, 10, 15, 20 of subculture for 20 generations on PDA slants. Wang et al. [3] also confirmed that the thermo-tolerant Lentinula edodes strain N44 by UV-induced protoplast mutagenesis is very stable in the fruiting experiments after successive subcultures for 10 generations. Under comprehensive consideration, subcultures for 15 generations were used to verify their stability in the present study.

[1]   Li Gang, Yang Fan, Li Ruixue, et al. A Study on the breeding of new Ganoderma strains by UV induced mutagenesis. Acta Microbiologica Sinica, 2001, (02): 229-233.

[2]   Han Yejun, Cao Hui, Chen Mingjie, et al. selection for cold-resistant strains by complex mutagenesi of Volvariella volvacea and identification of the strain. Acta Mycosystema, 2004, (03):417-422.

[3]   WANG LiNing, ZHAO Yan, ZHANG Bao-Fen, CHEN Ming-Jie. Breeding thermo-tolerant strains of Lentinula edodes by UV induced protoplast mutagenesis. Microbiol China, 2014, 41(7): 1350-1357.

Q2. Should the Figure caption be under the Figure? Please check the format again

Response: Thanks for your reminder. All figure captions have been revised in our revised Ms.

Q3. There are some places you forgot to make the scientific names italicized. Please check them again

Response: Thanks for your reminder. All scientific names have been modified and marked with red color in our revised Ms.

Reviewer 3 Report

The idea is novel however the title is quite confusing and should be direct

Ganoderma lucidum should be italicized throughout the whole manuscript

Graphical abstract is needed

UV treatments with pictures

Ultraviolet ray irradiation is an efficient and practical method for Ganoderma strain improvement and thus provided a basis for the development and application of G. lucidum spore production

...outstanding contributions to the rapid development of G. lucidum industry [this part should be added with more recent references] eg. https://doi.org/10.3390/su141710764

All tables should have significant symbols to compare between rows

Discussion should have a comparison table with recent Ganoderma cultivars

Justify why using ultraviolet irradiation to boost G. lucidum spore output is a good idea. Mycelial properties, resistance to other microbes, and spore productivity were among the criteria used to screen and ultimately choose the UV119 strain. It has dense, tiled mycelia. This mushroom has a thick mycelial layer and can produce a good spawn ratio of up to 100% in controlled settings. It produces 8.67% more fruiting bodies and 19.27% more basidiospores than the original strain G0109, with the latter figure being particularly impressive at 20.56% more than the current primary cultivar "Longzhi No.1." To be clear, UV119 outperforms the latter in both small and large-scale field agriculture. This means it will soon gain popularity and find widespread use.

Author Response

Dear Reviewer,

We would like to thank you for your efforts concerning our manuscript entitled “A novel strain breeding of Ganoderma lucidum UV119 (Agaricomycetes) with high spores yield and strong resistant ability to other microbes’ invasions” (foods-2070359). Those comments are all valuable and very helpful for revising and improving our paper, as well as the important guiding significance to our researches. We have studied comments carefully and have made correction which we hope meet with approval. Revised portion are marked in red in the paper. The main corrections in the paper and the responses to the reviewer’ comments are as follows:

Reviewer(s)' Comments to Author:

Reviewer: 3

Q1. The idea is novel however the title is quite confusing and should be direct.

Response: Thanks for your comments. We have revised the title of our Ms to “A novel strain breeding of Ganoderma lucidum UV119 (Agaricomycetes) with high spores yield and strong resistant ability to other microbes’ invasions”.

Q2. Ganoderma lucidum should be italicized throughout the whole manuscript.

Response: Thanks for your reminder. We have modified the above errors in our revised Ms.

Q3. Graphical abstract is needed.

Response: Thanks for your comments. We have supplemented the Graphical abstract in our revised Ms.

Q4. UV treatments with pictures

Response: Thanks for your comments. According to your request, the protocol for UV treatment was as follows.

Q5. Ultraviolet ray irradiation is an efficient and practical method for Ganoderma strain improvement and thus provided a basis for the development and application of G. lucidum spore production ...outstanding contributions to the rapid development of G. lucidum industry [this part should be added with more recent references].

Response: Thanks for your reminder. We have added some references according to your suggestion in our revised Ms.

Q6. All tables should have significant symbols to compare between rows.

Response: Thanks for your reminder. All tables have been modified according to your suggestion in our revised Ms.

Q7. Discussion should have a comparison table with recent Ganoderma cultivars

Response: Thanks for your comments. We have supplemented the comparison table with recent Ganoderma cultivars in our revised Ms.

Q8. Justify why using ultraviolet irradiation to boost G. lucidum spore output is a good idea. Mycelial properties, resistance to other microbes, and spore productivity were among the criteria used to screen and ultimately choose the UV119 strain. It has dense, tiled mycelia. This mushroom has a thick mycelial layer and can produce a good spawn ratio of up to 100% in controlled settings. It produces 8.67% more fruiting bodies and 19.27% more basidiospores than the original strain G0109, with the latter figure being particularly impressive at 20.56% more than the current primary cultivar "Longzhi No.1." To be clear, UV119 outperforms the latter in both small and large-scale field agriculture. This means it will soon gain popularity and find widespread use.

Response: Thanks for your professional review work on our Ms. Artificial mutagenesis using UV irradiation has also been conducted to change basidiospores yield of some mushroom species, including Pleurotus pulmonarius [1] and Pleurotus ostreatus [2]. In this respect, UV irradiation may be used to alter the yield of G. lucidum. Moreover, according to the study [1], an msh4 homolog, stpp1, from Pleurotus pulmonarius, is related to the production of the spores, its 193 bp length region was deleted in the sporeless mutant strain. The sporeless trait was temperature independent and had no linkage to unfavorable traits affecting fruiting, spore germination, or mycelial growth. There, the msh4 homolog from G. lucidum, is cloned and sequenced. Two little fragments (22 bp, 20 bp) deletion is found through an alignment using Mega 7.0 software. Maybe the deletion is relevant to the increase in basidiospore yield. The result wasn’t presented in this present study.

Sequences of msh4 of G. lucidum were as follows.

begin characters;

       dimensions nchar=3372;

       format missing=? gap=- matchchar=. datatype=nucleotide;

       matrix

MT(UV119):

ATGCATCTCCACTATCCTTCAATCGTCCTTGTTCCTGACACGTTCATGTCCCTGTCTGATGTTTCGCTCCCATCTGGTGCGAAGACCCCGCAGACGACAACACTTCTCGTTCAATGCATCATGGACGAGTTCGACGGTGTACCCATCGAGCCAGTCATGCGCAAATATTGGAGCGATAACGCAGGTGAGTACTCGGGTCTCTCGCTTGCGTGCACAGCACTGATAAATACATTAGGGCTAGACTTCATCAACCAGCTCATGGTCGACAACGACGAACGTGCTGCCACCCTCGTCGCCGTGTCCAATAAGTGCATATATGCACCTAAACTCTAGCATCTCTTGCGTGCTCAACATTGCCACAGATACTATGCACTCTCTGCCGCCAGCGCGCTCTTCAAGCATGCTGAACTCAAGCTCAACGTGCGCTTTGCTGCTTCGTCTCTACTTATCCGATACACCCAGGTGGAAGGTACAATGATGATTGACTCTGACACCGCTCGCAATCTCGAGCTGGTTGGCAACATGTCAGTTCGGAAGAGTGCGCATTCTCTTTTCGCGTACGTTCATATGTCTTCATTTAGCTCAAGCTTGGTGGTGACTCAGATATATGCGCAGTTTGTTGAACCATACGTATACGGCCATGGGCGCAAGACTGTTGAGGGTCAACATTTTAGCTCCTATCACTGGTCAGTCTCGTGCAACAGCGTTCGTTGAACCCAAGAGTTAACATATGCATATGCTGCCAGTGAAATCTGCACTGGAAGCACGACTAGATGTAGTCGAAGGTACAATCATCTTGCTGTCAGTGTATAAGTATCTAATCCAAATCTAATCACGTCTCGACTAGAGTTCATCCAAACTGAAGACCGTTTCACGGACGTCAAGGACGCTCTCAAATCACTTAACAAGCTCGATTTCGACAAGCTTATTTGTTCTGTGCGTCTCTTCAAGGCCACCGCTTCAAGGACACTGTGCTGGGAGATATTCATGTCGAATGCAGCTCGCATCCTCCGAAGTTCGAGAAGTTAGCACCTCAAAAATAGCATCGGCGAGAGTCGCACAAATGCTCAACCTCAGGAACATTGTCCAAAGTCTCCCCCGGCTTGCCAAGGCTCTTGAGGGAAGTCGGTCGCAGCTGTTACAAATCATCGCAGAGGTTCGCCTTAGTCTGTATCTCGAGTACCACGGCTGTGCTTAGTGCCAAGGACTATCCTCAGATGATATCAGACGATCGTCTGGAGAAGATCGAGCGGTTGGTATGCGACCGGTTGAATGAGGAGACAACTCCCGCAAAGGTAATCGTCATGCAATGCCATGATTATCGAGACTCTTAACATCCATATCAAGGGAGGGTTGAATGCGGTGAACGCTAGAGTCTACGCAGTTAAGGTAATGTCTGCATTGTCTGCCGAGGACTTGATTGAACGAATCTTCCATCAGGCTAATTTCAATCGCCTGCTCGACGTTGCTCGTGAAACCTATAAGGAGAACGTCGGCGACATCTATGCTCTGCGCAATCGTCTTGCCGAAGAACACAACCTCCCTTTCACTCTAGTGTACCGCGATTCTGATGCGGGCTTTGTCTTCTATATCAAGAAGACTGACCTTGACGAAGCTGGCGGCGAGCTTCCTCGCGGCTTCATCGACGCGATGGCACAGAAAGGAAGATGGGTGTTCTCGAGTATCGAGCTGGTGAGGAGGATATTCCCTATGTACATCGGCAAGAAGAGCCTGAACATCAACAGAAAAAGAGAAACGCACGCATGAAGGATGCGCTGGACGAGTCCTTGATCCTTAGTGACAAGCGAGTACATCGGATTTCTAGTGGCTTACATCGGTCTGAAGTGCGTCTCAAGGATCATTCAGGACCTCACAGACGAGATAGTCGTCGATATGGGTGCTCTGTACAAGGCATCCGAAGCAGTAGCGCTCCTTGATCTCTTATGGT----------------------CATTTGCGCATGCCTCAATCCGTGAGCCCTTCGTTTGTTGTGTTTTCACCCCGCGCTTACAAATTTTAGTGCGAAACTATGGTTCGTCCCTGAGTACTCCCCCGCGATGCCTTCCCGAAGGCTCATGTAGCTGTGCTGTGCGACAGTGCGCCCGGAGTTCACTGGGACTCTTGCAATCAAAGGTGGCCGCCATCCGGTTCTGGAATGTGTACAGTCTGCCGGCGCGACTATCCCCAACGACGTTTATTGTTGCGAGGCGTCTTCATTCCAGATTGTTCAAGGTCCAAAGTACGTGTCCCGTACTGGTTGGTTTTCCTACGCCACTGGGATATTGAGCTGGTTTGCCCCAGCATGTCGGGCAAAAGCACGTATCTACGACAGATTGCCCTGATAACAGTGATGGCAATGTGTGGTTGTTTCATTCCCGCAGAGTATGGATCATTCAGGTACACTCTAATGTGCTAACCTCTGTGGTTTGTTTTGAACAACGCGCCGACATATCCAGGATTCATGATCATCTGCTTACCCGACTTTCGAATGATGACGATCTGGAGAAGAACCTGAGCACGTTCGCGAATGAAATGGCCTCAACTGCCATGATTCTAGGTACATTTCAACTATGTCCCGTGCGCACCTCTCTCATTCTGACTAGGTCTGGCCACTGAGAACTCCCTGGTACTCATCGATGAAGTGGGTCGGGGTACCTCGGCGAGAGAAGGCGTAGCTATCTCACATGCCATTGCGGAGGAGCTCAT--------------------GGTCCCTTCATCTTATAAGCCCACTCTGTCTAACCATTCAAACCAGTCCTTTGTCTTCTTTGCAACGCAAGGGTCCACTACAATTCTAACCCGTCAGCCTTTCACTGATGGTCAGATCTCAGACATTTCAACAAGCTCACCACTACACTCTCCCGACAGCCATCTGTCGTCAAGTAAACTTTTACCGGTGGCTTCCTCCAAGGGTGGGTACTGACGCCGTCCCCTACAGTTTACATCTATCCTTCCAGGTACTCTTTTCCTATAACCTGCCTGTTTTGTGAACTACGTGTTCAACAAATTACAGAAATCACTGCCGACAGCCTCTAAAGTTGGCATCACTTTCCACTACAAGTAAGAAGCTGTCTCTTTATCCGCTCTAAGCCTCTCCTCATCAGCCCTTAGGATTATGGATGGCGCTCCGGAAACTCAAGACCATTATGGTATGATCGTTACCTATTGAACGCTCACATACTGCCAAACAGTCGTCTAGGCTTAGACCTTGCTCGTTTGGCGGACCTCCCAGAGTCCGTTGTATCCGAGGCTCGACGTGTCGCTGAGTACCTCGCGAAACAAGAGGAGCGGGACCAGCAGCAGAGTAAGACGAGCAAGATTGCGCTCCGCCGCAAA

WT(G0109):

ATGCATCTCCACTATCCTTCAATCGTCCTTGTTCCTGACACGTTCATGTCCCTGTCTGATGTTTCGCTCCCATCTGGTGCGAAGACCCCGCAGACGACAACACTTCTCGTTCAATGCATCATGGACGAGTTCGACGGTGTACCCATCGAGCCAGTCATGCGCAAATATTGGAGCGATAACGCAGGTGAGTACTCGGGTCTCTCGCTTGCGTGCACAGCACTGATAAATACATTAGGGCTAGACTTCATCAACCAGCTCATGGTCGACAACGACGAACGTGCTGCCACCCTCGTCGCCGTGTCCAATAAGTGCATATATGCACCTAAACTCTAGCATCTCTTGCGTGCTCAACATTGCCACAGATACTATGCACTCTCTGCCGCCAGCGCGCTCTTCAAGCATGCTGAACTCAAGCTCAACGTGCGCTTTGCTGCTTCGTCTCTACTTATCCGATACACCCAGGTGGAAGGTACAATGATGATTGGCTCTGACACCGCTCGCAATCTCGAGCTGGTTGGCAACATGTCAGTTCGGAAGAGTGCGCATTCTCTTTTCGCGTACGTTCATATGTCTTCATTTAGCTCAAGCTTGGTGGTGACTCAGATATATGCGCAGTTTGTTGAACCATACGTATACGGCCATGGGCGCAAGACTGTTGAGGGCCAACATTTTAGCTCCTATCACTGGTCAGTCTCGTGCAACAGCGTTCGTTGAACCCAAGAGTTAACATATGCATATGCTGCCAGTGAAATCTGCACTGGAAGCACGACTAGATGTAGTCGAAGGTACAATCATCTTGCTGTCAGTGTATAAGTATCTAATCCAAATCTAATCACGTCTCGACTAGAGTTCATCCAAACTGAAGACCGTTTCACGGACGTCAAGGACGCTCTCAAATCACTTAACAAGCTCGATTTCGACAAGCTTATTTGTTCTGTGCGTCTCTTCAAGGCCACCGCTTCAAGGACACTGTGCTGGGAGATATTCATGTCGAATGCAGCTCGCATCCTCCGAAGTTCGAGAAGTTAGCACCTCAAAAATAGCATCGGCGAGAGTCGCACAAATGCTCAACCTCAGGAACATTGTCCAAAGTCTCCCCCGGCTTGCCAAGGCTCTTGAGGGAAGTCGGTCGCAGCTGTTACAAATCATCGCAGAGGTTCGCCTTAGTCTGTATCTCGAGTACCACGGCTGTGCTTAGTGCCAAGGACTATCCTCAGATGATATCAGACGATCGTCTGGAGAAGATCGAGCGGTTGGTATGCGACCGGTTGAATGAGGAGACAACTCCCGCAAAGGTAATCGTCATGCAATGCCATGATTATCGAGACTCTTAACATCCATATCAAGGGAGGGTTGAATGCGGTGAACGCTAGAGTCTACGCAGTTAAGGTAATGTCTGCATTGTCTGCCGAGGACTTGATTGAACGAATCTTCCATCAGGCTAATTTCAATCGCCTGCTCGACGTTGCTCGTGAAACCTATAAGGAGAACGTCGGCGACATCTATGCTCTGCGCAATCGTCTTGCCGAAGAACACAACCTCCCTTTCACTCTAGTGTACCGCGATTCTGATGCGGGCTTTGTCTTCTATATCAAGAAGACTGACCTTGACGAAGCTGGCGGCGAGCTTCCTCGCGGCTTCATCGACGCGATGGCACAGAGAGGAAGATGGGTGTTCTCGAGTATCGAGCTGGTGAGGAGGATATTCCCTATGTACATCGGCAAGAAGAGCCTGAACATCAACAGAAAAAGAGAAACGCACGCATGAAGGATGCGCTGGACGAGTCCTTGATCCTTAGTGACAAGCGAGTACATCGGATTTCTAGTGGCTTTCATCGGTCTGAAGTGCGTCTCAAGGATCATTCAGGACCTCACAGACGAGATAGTCGTCGATATGGGTGCTCTGTACAAGGCATCCGAAGCAGTAGCGCTCCTTGATCTCTTATGGTGCGCTCCTTGATCTCTTATGGTCATTTGCGCATGCCTCAATCCGTGAGCCCTTCGTTTGTTGTGTTTTCACCCCGCGCTTACAAATTTTAGT

GCGAAACTATGGTTCGTCCCTGAGTACTCCCCCGCGATGCCTTCCCGAAGGCTCATGTAGCTGTGCTGTGCGACAGTGCGCCCGGAGTTCACTGGGGCTCTTGCAATCAAAGGTGGCCGCCATCCGGTTCTGGAATGTGTACAGTCTGCCGGCGCGACTATCCCCAACGACGTTTATTGTTGCGAGGCGTCTTCATTCCAGATTGTTCAAGGTCCAAAGTACGTGTCCCGTACTGGTTGGTTTTCCTACGCCACTGGGATATTGAGCTGGTTTGCCCCAGCATGTCGGGCAAAAGCACGTATCTACGACAGATTGCCCTGATAACAGTGATGGCAATGTGTGGTTGTTTCATTCCCGCAGAGTATGGATCATTCAGGTACACTCTAATGTGCTAACCTCTGTGGTTTGTTTTGAACAACGCGCCGACATATCCAGGATTCATGATCATCTGCTTACCCGACTTTCGAATGATGACGATCTGGAGAAGAACCTGAGCACGTTCGCGAATGAAATGGCCTCAACTGCCATGATTCTAGGTACATTTCAACTATGTCCCGTGCGCACCTCTCTCATTCTGACTAGGTCTGGCCACTGAGAACTCCCTGGTACTCATCGATGAAGTGGGTCGGGGTACCTCGGCGAGAGAAGGCGTAGCTATCTCACATGCCATTGCGGAGGAGCTCATCCGACTGAAAGTACGGACATGGTCCCTTCATCTTATAAGTCCACTCTGTCTAACCATTCAAACCAGTCCTTTGTCTTCTTTGCAACGCAAGGGTCCACTACAATTCTAACCCGTCAGCCTTTCACTGATGGTCAGATCTCAGACATTTCAACAAGCTCACCACTACACTCTCCCGACAGCCATCTGTCGTCAAGTAAACTTTTACCGGTGGCTTCCTCCAAGGGTGGGTACTGACGCCGTCCCCTACAGTTTACATCTATCCTTCCAGGTACTCTTTTCCTATAACCTGCCTGTTTTGTGAACTACGTGTTCAACAAATTACAGAAATCACTGCCGACAGCCTCTAAAGTTGGCATCACTTTCCACTACAAGTAAGAAGCTGTCTCTTTATCCGCTCTAAGCCTCTCCTCATCAGCCCTTAGGATTATGGATGGCGCTCCGGAAACTCAAGACCATTATGGTATGATCGTTACCTATTGAACGCTCACATACTGCCAAACAGTCGTCTAGGCTTAGACCTTGCTCGTTTGGCGGACCTCCCAGAGTCCGTTGTATCCGAGGCTCGACGTGTCGCTGAGTACCTCGCGAAACAAGAGGAGCGGGACCAGCAGCAGAGTAAGACGAGCAAGATTGCGCTCCGCCGCAAA

[1]   Okuda Y, Murakami S, Honda Y, et al. An MSH4 Homolog, stpp1, from Pleurotus pulmonarius is a "Silver Bullet" for resolving problems caused by spores in cultivated mushrooms. Applied & Environmental Microbiology, 2013, 79(15): 4520-4527.

[2]   Pandey M, Ravishankar S. Development of sporeless and low-spored mutants of edible mushroom for alleviating respiratory allergies. Current Science, 2010, 99(10):1449-1453.

Round 2

Reviewer 1 Report

1. Q2 was answered unsatisfactorily. The reviewer needs the data on how many cycles the substrate could possibly be used again after the 1st fruiting and a comparison has to be made with the control mycelium (without mutated gen). This would possibly be the main cause of how efficiently the substrate and mutated cells could affect the cycle growth (recyclability). 

2. Q4: I don't understand the response given and I'd consider it vague and irresponsive to the question given. 

3. Final verdict: Rejected 

Author Response

Dear Reviewer,

First of all, we would like to thank you for your efforts concerning our Ms entitled “A novel strain breeding of Ganoderma lucidum UV119 (Agaricomycetes) with high spores yield and strong resistant ability to other microbes’ invasions” (foods-2070359). Through the first round of polishing, our Ms has been improved to a higher level. In response to your comments (R2), we have now made a serious reply, hoping to meet with approval. The responses to the reviewer’ comments are as follows:

Question 1: Q2 was answered unsatisfactorily. The reviewer needs the data on how many cycles the substrate could possibly be used again after the 1st fruiting and a comparison has to be made with the control mycelium (without mutated gen). This would possibly be the main cause of how efficiently the substrate and mutated cells could affect the cycle growth (recyclability).

Response: Thanks for your comments.

For a bag culture, if it is only used to produce fruiting bodies and we generally harvest the first flush fruiting bodies when the Ganoderma lucidum fruiting bodies are developmental mature and have not yet released basidiospores. And then the second flush fruiting bodies begin developing when the temperature, humidity, CO2 concentration and light in cultivation room are suitable for G. lucidum development. Generally, we use 1kg of wet substrate composed of 78% of sawdust, 13% of wheat bran, 7% of corn powder, 1% of sucrose and 1% of gypsum by dry weight, with water content at 65%, in cultivation of G. lucidum.

Here, a comparison of the mutant strain UV119 with the originating strain G0109 was carried out in cultivation. We found that both of them formed primordia almost at the same time, and their fruiting bodies developed almost at the same pace and were harvested at 40-45 days after the formation of primordia. The dry weight of fruiting bodies of UV119 was 20.72±0.37 g/spawn, and that of G0109 was 19.63±0.55 g/spawn. The second flush fruiting bodies both stopped growing on their 36-38 days’ developing after harvesting of the first flush fruiting bodies. The dry weight of the second flush fruiting bodies of the above two varieties was 10.48±0.63 g/spawn and 8.43±0.53 g/spawn, respectively. For the two varieties, the residual substrate couldn’t produce the third flush fruiting bodies, which the nutrition of the substrate was almost exhausted after two flushes, even if the condition of the temperature, humidity, CO2 concentration and light was very suitable for fruiting bodies. We could conclude that the yield of the second fruiting bodies was almost half that of the first flush.

Actually, because the price of basidiospores powder is much higher than that of G. lucidum fruiting bodies, for the varieties with high G. lucidum spore powder, such as UV119 and G0109, all growers generally try to make it produce as much spore powder as possible. When G. lucidum fruiting bodies start to release spore powder, they will begin to collect spore powder using non-woven cylinders lined with food preservation bags or other means. In the course of spores collection, the temperature was controlled at 26-28℃ and the humidity was adjusted to 80% or so. The basidiospores releasing lasts about 35-40 days. The spore yield of the two varieties is 19.50±0.36 g/spawn and 16.25±0.40 g/spawn, respectively. And the nutrients in the substrate have been almost exhausted.

In the field cultivation, the growers use the wood packed with polypropylene bags as the substrates. The wood weight is 10kg/ bag. The nutrients in the wood are only partially consumed by the G. lucidum fruiting bodies in one year (one growth cycle). So, if there is no disease, the second flush G. lucidum fruiting bodies can grow on this wood in the next year (the second growth cycle). According to our statistics in the recent two years, the fruiting bodies production rates of varieties UV119 and LZ-1 in the first year were 99.50% and 90.13%, respectively, and the spores powder yield in the first year was 0.019 kg/ wood and 0.015 kg/ wood, respectively. In the second year, the fruiting rate of UV119 reached about 90.00%, while the fruiting rate of LZ-1 was only about 30.00% due to its weak resistance to disease. Their spores powder yield was 0.01 kg/ wood and 0.001 kg/wood, respectively. After our detection, the nutrition of the wood, on which still growing G. lucidum fruiting body in the second year, has been completely consumed.

Question 2: Q4: I don't understand the response given and I'd consider it vague and irresponsive to the question given. (I don't see any conclusive evidence to confirm that the irradiation process would possibly alter the gene at the molecular level of the used spores. It was just a physical effect in terms of its yield and none of the confirmation was presented in the main text. Changes to the strain need to be carried out via physic-chemical analysis or perhaps details molecular level profiles to confirm it definitely.)

Response: Thanks for your comments.

It is common knowledge that ultraviolet irradiation causes genetic mutations. Some studies have reported that ultraviolet irradiation can change the production of fungal basidiospores [19, 20, 30, 31]. In this study, the protoplasts of G0109 were treated by ultraviolet irradiation, and a mutant strain UV119 with significantly increased spores powder yield was obtained. The results also showed that the strain had mutation. G. lucidum spores are the sexual basidispores released by the matured pileus and also its product of meiosis. We never describe the molecular changes of the genes of basidiospores caused by UV irradiation.

References:

  1. Okuda, Y.; Murakami, S.; Honda, Y.; Matsumoto, T. An MSH4 Homolog, stpp1, from Pleurotus pulmonarius is a "Silver Bullet" for resolving problems caused by spores in cultivated mushrooms. Appl Environ Microb 2013, 79(15), 4520-4527.
  2. Pandey, M.; Ravishankar, S. Development of sporeless and low-spored mutants of edible mushroom for alleviating respiratory allergies. Curr Sci India 2010, 99(10), 1449-1453.
  3. Hadibarata, T.; Kristanti, R.A. Potential of a white-rot fungus Pleurotus eryngii F032 for degradation and transformation of fluorene. Fungal Biol 2014, 118, 222-7.
  4. Zolan, M.E.; Tremel, C.J.; Pukkila, P.J. Production and characterization of radiation-sensitive meiotic mutants of Coprinus cinereus. Genetics 1988, 120(2), 379-87.
